# Assessment of the prognostic value of CA125 for miscarriage risk in patients with threatened abortion: A systematic review and meta-analysis

Yurong Cao[1]*, Weiwei Li[1], Linna Ma[2]

1 Department of Obstetrics and Gynecology, Union Hospital, Tongji Medical College, Huazhong University of Science and Technology, Wuhan, China, 2 Department of Reproductive Medicine, the First Affiliated Hospital of Hainan Medical College, Hainan, China

* cyrzkx2023@163.com

## Abstract

### Background

Threatened abortion is a common obstetric complication where early risk stratification is crucial for clinical management. CA125, a glycoprotein biomarker produced by endometrial and trophoblastic tissues, may serve as a valuable prognostic indicator as its elevated levels often reflect placental inflammation or detachment in this condition. To systematically assess its clinical value, we conducted a meta-analysis evaluating the prognostic accuracy of serum CA125 levels for miscarriage risk in patients with threatened abortion.

### Methods

We systematically searched Web of Science, Medline via PubMed, ScienceDirect, EMBASE, Geenmedical, Cochrane Library, and Wiley databases from inception to May 30, 2024 for studies evaluating either CA125's prognostic value for miscarriage risk in threatened abortion, or serum markers' prognostic value for pregnancy outcomes. Two researchers independently screened literature and extracted data. Study quality and evidence certainty were assessed using QUAPAS and GRADE frameworks respectively. Meta-analyses were performed using Stata 17 and Review Manager 5.3.

### Results

A total of 13 studies involving 1166 patients with threatened abortion were included in this meta-analysis. The pooled estimates demonstrated that CA125 exhibited the following prognostic performance for predicting miscarriage risk in patients with threatened abortion: sensitivity, 89% (95% CI: 83–93%); specificity, 91% (95% CI: 84–95%); positive likelihood ratio (PLR), 10.2 (95% CI: 5.4–19.3); negative likelihood

**Data availability statement:** All relevant data are within the paper and its Supporting Information files.

**Funding:** The author(s) received no specific funding for this work.

**Competing interests:** The authors have declared that no competing interests exist.

ratio (NLR), 0.13 (95% CI: 0.08–0.19); diagnostic odds ratio (DOR), 82 (95% CI: 31–212); and area under the summary receiver operating characteristic curve (SROC AUC), 0.95 (95% CI: 0.92–0.96). Elevated serum CA125 levels were significantly associated with an increased risk of miscarriage. According to the GRADE assessment, the certainty of evidence was moderate for sensitivity but low for other parameters, primarily due to heterogeneity, risk of bias, and other confounding factors.

## Conclusion

Current evidence with moderate certainty indicates that serum CA125 demonstrates high prognostic value for assessing miscarriage risk in threatened abortion, particularly for identifying high-risk patients. However, due to low-certainty specificity findings, caution is warranted when using CA125 to rule out miscarriage. While these results support CA125's potential as a complementary tool to ultrasound, further studies are required to establish standardized cut-off values for clinical implementation.

## Introduction

Threatened abortion, a frequent obstetric emergency occurring before 20 weeks of gestation, is clinically characterized by vaginal bleeding and/or abdominal pain in the presence of a closed cervical os and demonstrable fetal cardiac activity on ultrasonography. Epidemiological data indicate that 15%−20% of pregnancies complicated by threatened abortion ultimately progress to miscarriage [1–3]. Threatened abortion imposes significant physical, psychological, and socioeconomic burdens on affected women and their families, while potentially impacting prognosis [4]. Consequently, precise risk stratification for miscarriage in these patients is crucial for optimal clinical management. While transvaginal ultrasonography and serial β-human chorionic gonadotropin (β-hCG) measurements currently represent the standard prognostic modalities [5,6], these conventional biomarkers demonstrate limited predictive accuracy for miscarriage risk.

In recent years, increasing research on biomarkers has shown that changes in specific protein levels may be closely related to pregnancy outcomes. Among these biomarkers, carbohydrate antigen 125 (CA125), a glycoprotein physiologically expressed by coelomic epithelial cells, has garnered significant research interest. During pregnancy, CA125 is produced by decidual cells and remains primarily localized within the amniotic cavity. However, when placental barrier integrity is compromised in threatened abortion, CA125 enters maternal circulation, where its levels correlate with the extent of trophoblast injury [7–10]. This mechanistic link supports its potential as a prognostic biomarker, though its clinical value for miscarriage risk prediction remains controversial with inconsistent results across studies.

Therefore, in this paper, based on the existing literature, we will use meta-analysis to comprehensively evaluate the clinical application value of CA125 level in predicting miscarriage risk in patients with threatened abortion with a view to providing a reference basis for clinical practice.

## Materials and methods

### Inclusion and exclusion criteria

Inclusion Criteria: 1) Articles should include patients with threatened abortion. Threatened abortion was defined according to American College of Obstetricians and Gynecologists guidelines [11]: 5-20w gestation, diagnosed as intrauterine live birth with threatened abortion by ultrasound, the diagnostic criteria of threatened abortion were the presence of small amount of vaginal bleeding, paroxysmal lower abdominal pain or lumbar pain, gynecological examination of the uterine orifice was not open, fetal membranes were intact, there was no gestational expulsion, and uterine size was in line with the week of gestation; 2) the type of the study was a prospective or retrospective trial and the topic of study was the relationship between the markers and the pregnancy outcome, and the marker was or included blood CA125; 3) Sensitivity and specificity values were available; 4) Relevant literature in English, regardless of age, country, or race.

Exclusion criteria: 1) case reports, conference abstracts, systematic evaluations, reviews, animal experiments, letters; 2) duplicate publications with incomplete or contradictory data.

Outcome indicators: sensitivity (SEN), specificity (SPE), positive likelihood ratio (PLR), negative likelihood ratio (NLR), diagnostic odds ratio (DOR), and area under curve (AUC) of summary receiver operating characteristic (SROC).

### Databases and search strategies

This study followed the PRISMA guidelines [12] for systematic evaluation and meta-analysis and was registered in the International prospective register of systematic reviews (PROSPERO, CRD42024600834).

The databases Web of science, Medline via PubMed, Sciencedirect, EMBASE, Geenmedical, Cochrane Library, and Wiley were searched to find the relevant literature, and the search date was from the construction of the library to May 2024, and the search was conducted by the search terms of medical topics: (serum cancer antigen125)OR(serum CA125) OR(CA125)OR(CA-125)and(threatened abortion)OR(threatened miscarriage)OR(pregnancy loss)OR(first trimester abortion)OR(first trimester pregnancy)OR(early pregnancy failure)OR(pregnancy outcome).

Literature search, screening and data extraction were performed independently by two researchers, and the results were cross-checked, with a third researcher being asked to arbitrate in case of disagreement, in addition to which the researchers screened the reference lists of relevant reviews and articles to ensure that the search was comprehensive and free of omissions.

### Data extraction and quality assessment

Two researchers systematically extracted the following raw data from each included study: first author, year of publication, country, type of study, inclusion and exclusion criteria, sample size, TP, FP, TN, FN, SEN, SPE, cut-off value, diagnosis of miscarriage and follow-up deadlines. In particular, when TP, FP, TN, FN, were not directly reported, we calculated them using the reported SEN and SPE values along with the sample size. The article used the Quality Assessment of Prognostic Accuracy Studies (QUAPAS) risk of bias assessment tool to assess the risk of bias and applicability of the included studies. QUAPAS consists of 5 main aspects: participants bias, index test bias, outcome bias, flow and timing bias, and analysis method [13]. The index test for inclusion in the study was CA125, and the outcome determination was miscarriage confirmed by clinical or USS or histopathologic examination during the follow-up period.

### Statistical analysis

Meta-analysis was performed using Review Manager 5.3 and Stata 17.0 software to plot the SROC plan, observe whether the curve showed a "shoulder-arm" distribution, and calculate the correlation coefficients between the logarithm of Sen and the logarithm of (1-Spe) Spearman to determine whether there was heterogeneity due to the threshold effect. The $\chi^2$ test or Cochran-q test was used to detect the presence of heterogeneity due to non-threshold effects, and a fixed-effects

model was used if there was no heterogeneity or less heterogeneity with $I^2 \leq 50\%$ and $P \geq 0.10$, and a random-effects model was used if heterogeneity existed with $I^2 > 50\%$ and $P < 0.10$. Prognosis was assessed by combining SEN, SPE, PLR, NLR, DOR and AUC. To investigate the potential sources of high heterogeneity unrelated to threshold effects, we conducted a meta-regression analysis. Deek's funnel plot was used to detect publication bias, and if the funnel plot was symmetrical, it indicated that there was no publication bias ($P > 0.05$). The raw statistics and packages used in STATA program are detailed in S1 File.

## Results

### Study selection

A total of 6149 literatures were obtained by searching the online database, 54 literatures were obtained after reading the title and deleting the duplicate articles, 21 were excluded after reading the abstract, and the remaining 33 articles were searched for the full text, of which 2 were not retrieved from the original literature, the remaining 31 were further extracted, 1 was a review, 8 were excluded due to the mismatch of the research objects, and the remaining 22 were retained. Sensitivity and specificity could not be obtained in 8 of them, and data inconsistencies existed in 1 of them. Finally, 13 literatures were included in the meta-analysis, of which 8 were prospective studies. The flow chart of article selection is shown in Fig 1. Every excluded article and the reasons for exclusion are detailed in S2 File.

### Study characteristics and risk of bias

A total of 13 articles [14–26] were included in this study, including 1166 patients with threatened abortion from Asia (1/13, 135patients), Europe (4/13, 718patients) and Africa (8/13, 313patients), including 387 patients with abortion and 779 patients with normal pregnancy or delivery after the follow-up period. The main characteristics of the included literatures are shown in Tables 1 and S1, and the results of article quality evaluation are shown in Fig 2. The assessment of applicability and judgment rationale behind the use of QUAPAS tool is detailed in S3 File.

### Meta-analysis results

**Heterogeneity test.** The SROC did not present a "shoulder-to-arm" distribution (Fig 3). Spearman correlation analysis suggested that the correlation coefficient was −0.588, $P = 0.044$, indicating no threshold effect. Studies were pooled despite varying CA125 cut-off values as threshold effect analysis demonstrated this was appropriate. Heterogeneity test was performed on all the included studies, and the results showed that the sensitivity ($\chi^2 = 27.54$, $P = 0.01$, $I^2 = 56.43\%$) and specificity ($\chi^2 = 66.00$, $P = 0.00$, $I^2 = 81.82\%$) among the results were highly heterogeneous, so the random effects model was used for meta-analysis.

**Combined effect size.** Meta-analysis showed that the sensitivity of CA125 for predicting miscarriage risk in patients with threatened abortion was 89% [95% CI (83−93%)], specificity was 91% [95% CI (84−95%)], Positive Likelihood Ratio (POR) was 10.2 [95% CI (5.4−19.3)], Negative Likelihood Ratio (NLR) was 0.13 [95% CI (0.08–0.19)], Diagnostic Odds Ratio (DOR) was 82 [95% CI (31−212)], Area Under SROC Curve was 0.95 [95% CI (0.92–0.96)], forest plots of specificity and sensitivity are shown in Fig 4, and the distribution of likelihood ratio quadrants is shown in Fig 5.

**Meta-regression.** In order to explore the source of heterogeneity, meta-regression analysis was performed on the covariates of the included references according to follow-up deadlines (≤20W, >20W), sample size (≤100, >100), and study population (Africa, other continents). The analysis revealed that follow-up deadlines did not significantly contribute to heterogeneity ($P = 0.11$). In contrast, both sample size ($P < 0.05$) and study population characteristics ($P < 0.05$) demonstrated statistically significant associations with observed heterogeneity. The observed heterogeneity in this study likely originates from disparities in sample sizes and population characteristics across included studies, potentially limiting the generalizable validity of the results. (Fig 6).

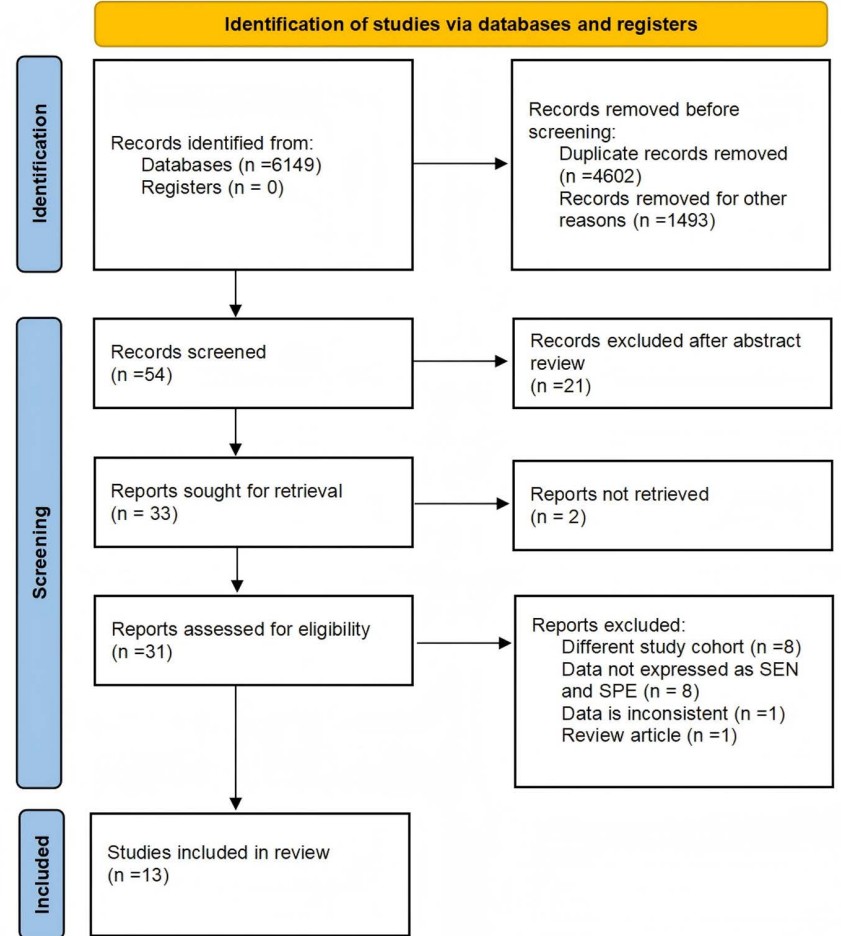

**Fig 1. Flowchart of study selection process for the systematic review and meta-analysis of CA125 in predicting miscarriage risk in patients with threatened abortion.**

**Publication bias.** Deek's funnel plot was drawn for publication bias test, and the results showed that the distribution around each study point was basically symmetric (P = 0.81), suggesting that there was a low possibility of publication bias (Fig 7).

**Summary of findings and GRADE certainty.** Using the GRADE (Grading of Recommendations, Assessment, Development, and Evaluations) framework, we assessed the certainty of evidence for each parameter. Sensitivity was rated as moderate certainty, downgraded due to risk of bias in patient selection and moderate heterogeneity. The specificity, PLR, NLR, DOR, and AUC were all rated as low certainty, primarily due to risk of bias in patient, moderate heterogeneity, imprecise confidence intervals and population imbalances. Threshold effect analysis (Spearman's $\rho = -0.588$, P = 0.044) confirmed that observed heterogeneity was not attributable to cut-off variability Table 2.

## Discussion

Threatened abortion represents a prevalent early pregnancy complication, occurring in 20%−25% of pregnancies. Among affected women, the miscarriage risk is 15.3%, while the preterm labor risk reaches 21%. Furthermore, this condition is associated with significantly increased incidences of fetal growth restriction, placenta previa, and pre-eclampsia during

**Table 1. Characteristics of the studies included in the systematic review.**

| Author(year) | Country/Continent | Study design | Study population | Sampling method | Utilized assay | Thresholds | Outcome measurement | Follow-up deadlines | Raw data |
|---|---|---|---|---|---|---|---|---|---|
| Fahri (1992) | Turkey/Europe | prospective study | 7-12W 25 threatened abortion | venipuncture | combined sandwich-solid phase radioimmunoassay | first : >65 U/mL after I-3 days: >60 U/mL | / | 20W | TP:5 FP:1 TN:19 FN:0 |
| Scarpellini (1995) | Italy/Europe | / | 6-11W 48threatened abortion | venipuncture | standard radioimmunoassay | 120IU/mL | USS | 24W | TP:7 FP:10 TN:29 FN:2 |
| Leylek (1997) | Turkey/Europe | / | 6-12W 40threatened abortion | / | / | 125IU/mL | / | end in abortion or delivery | TP:13 FP:1 TN:24 FN:2 |
| Sherif (2000) | Egypt/Africa | prospective study | 6-13W 100threatened abortion | venipuncture | chemiluminescent immunometric method | 21U/mL | USS | / | TP:41 FP:1 TN:56 FN:2 |
| Fiegler (2003) | Poland/Europe | / | 5-12W 200threatened abortion | venipuncture | enzyme immunoassay test | 43.1U/mL | hospital record | visit 4 weeks | TP:66 FP:18 TN:110 FN:6 |
| Maged (2013) | Egypt/Africa | prospective study | 5-12W 150threatened abortion | venipuncture | chemiluminescent immunometric method | 80IU/mL | / | end in abortion or delivery | TP:52 FP:19 TN:66 FN:13 |
| Xie (2014) | China/Asia | / | 5-12W 135threatened abortion | venipuncture | / | 55.57U/mL | USS or telephone interview | 28W | TP:72 FP:9 TN:47 FN:7 |
| Sweed (2016) | Egypt/Africa | prospective study | 6-12W 120threatened abortion | venipuncture | chemiluminescent immunometric method | 58IU/mL | USS | 20W | TP:23 FP:3 TN:87 FN:7 |
| Maged (2016) | Egypt/Africa | prospective study | 7-13W 100threatened abortion | venipuncture | chemiluminescent immunometric method | 31.2IU/mL | / | 20W | TP:19 FP:0 TN:80 FN:1 |
| Nesreen (2018) | Egypt/Africa | prospective study | 8-10W 100threatened abortion | venipuncture | enzyme immunoassay test | 31.8KU/L | assigned physician | 20W | TP:14 FP:16 TN:68 FN:2 |
| Mansy (2017) | Egypt/Africa | / | 5-13W 45threatened abortion | venipuncture | solid phase sandwich enzyme linked immunosorbent assay | 51.68U/ml | / | end in abortion or delivery | TP:8 FP:7 TN:29 FN:1 |
| Mosunmola (2019) | Nigeria/Africa | prospective study | 6-19+6W 63threatened abortion | venipuncture | / | 36.2IU/ml | antenatal records | delivery | TP:10 FP:8 TN:40 FN:5 |
| Mohamed (2020) | Egypt/Africa | prospective study | 6-12W 40threatened abortion | venipuncture | quantitative enzyme linked immunoassay | 30.3IU/ml | / | 20W | TP:9 FP:1 TN:30 FN:0 |

Note: FN: False Negative; FP: False Positive; TN: True Negative; TP: True Positive. USS: Ultrasound Scan System. All studies evaluated miscarriage as the primary outcome.

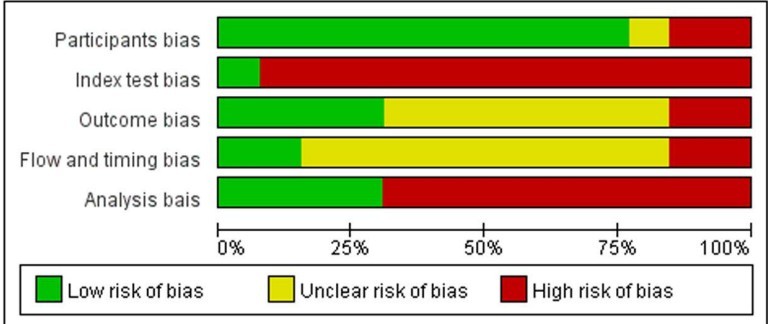

**Fig 2. Risk of bias assessment of included studies evaluating CA125 for miscarriage risk prediction in threatened abortion, analyzed using the QUAPAS tool.**

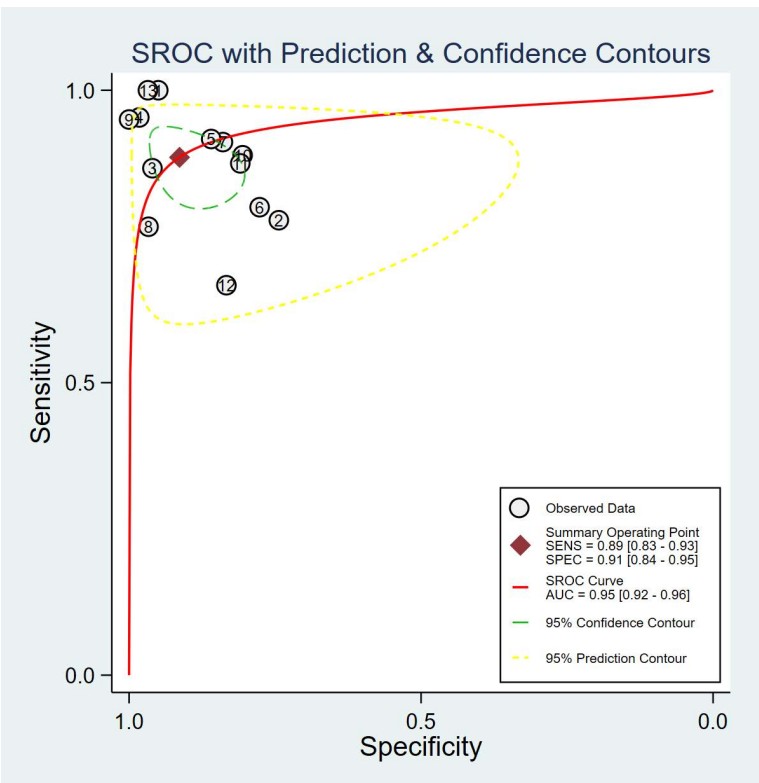

**Fig 3. Summary receiver operating characteristic (SROC) curve and empirical Bayes estimation of CA125 as a prognostic biomarker for miscarriage risk prediction in threatened abortion.**

later gestation [27–29]. Global statistics indicate approximately 23 million miscarriage cases occurred in 2021. Beyond physical complications such as hemorrhage and infection, miscarriage poses substantial psychological consequences, particularly for women who conceived through assisted reproductive technologies after significant financial and emotional investment. Severe cases may lead to clinical depression and even suicidal ideation [27,30]. Despite the various treatment options that have been developed, such as conservative and hormonal therapy, as well as various monitoring tools

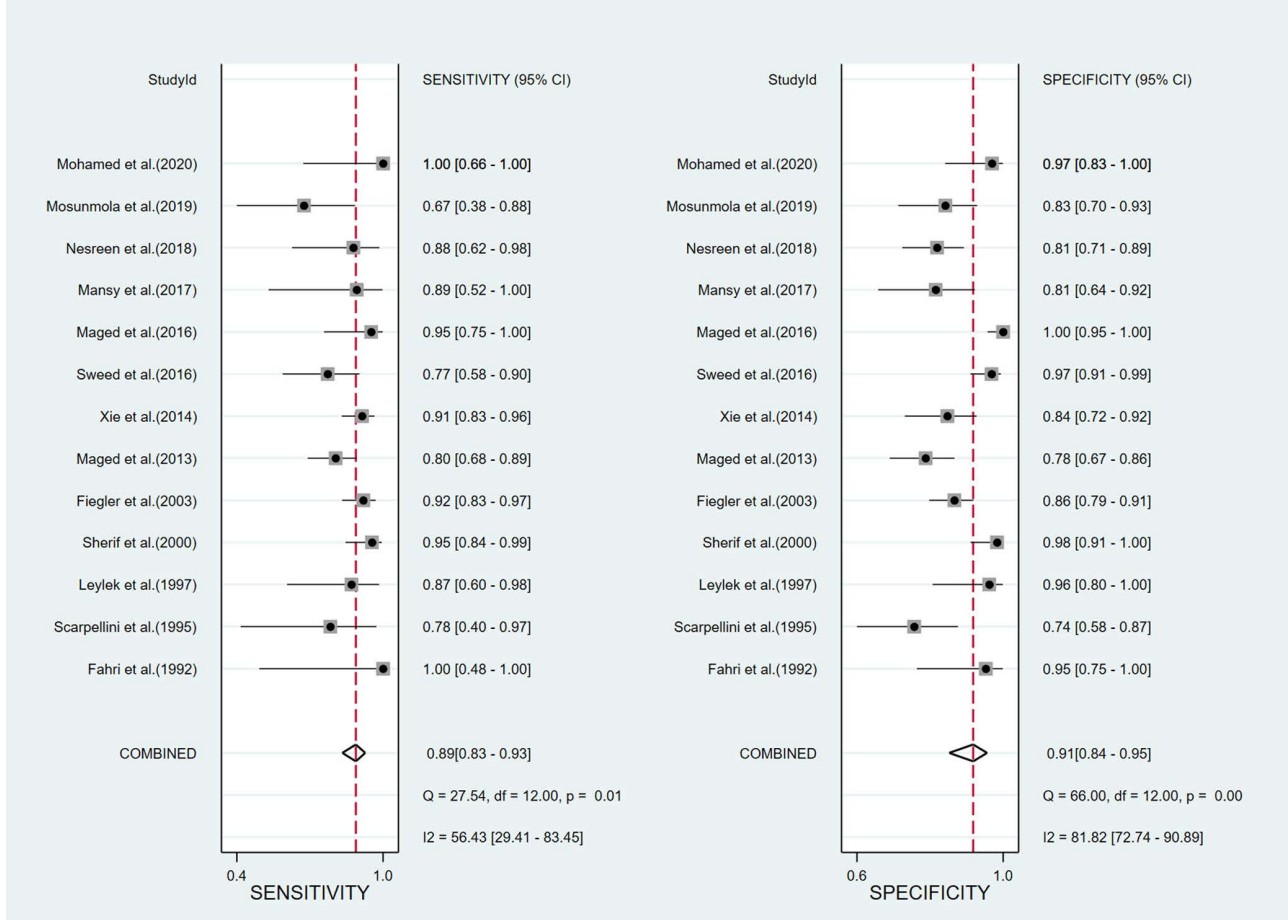

**Fig 4. Forest plot of meta-analysis evaluating the sensitivity and specificity of CA125 in predicting miscarriage risk in patients with threatened abortion.**

[31,32], threatened abortion or vaginal bleeding during pregnancy can still cause great psychological anxiety in patients, and the search for effective and accurate markers to predict miscarriage risk is crucial for clinical decision-making [33].

Our findings suggest that CA125 exhibits promising sensitivity (89%, moderate certainty) and specificity (91%, low certainty) for identifying high miscarriage risk in threatened abortion. However, the low-certainty evidence for PLR, DOR, and AUC indicates that its predictive values may vary across clinical settings. While the NLR (0.13, low certainty) hints at potential utility as a 'rule-out' tool in resource-limited areas, these results require validation through standardized studies before clinical implementation, as ultrasound remains the gold standard. And the results were similar to those of Pillai et al [34].

CA125 is a high-molecular-weight glycoprotein antigen expressed by coelomic epithelium-derived tissues, including the endometrium, peritoneum, and fallopian tube epithelium [35–37]. In healthy reproductive-age women, serum CA125 levels typically remain below 35 U/mL. However, its expression can be altered by hormonal fluctuations and various pathological conditions. Clinically, CA125 serves as an established biomarker for diagnosing and monitoring several gynecological malignancies, particularly ovarian epithelial carcinoma, as well as endometriosis and endometrial carcinoma, with additional utility in treatment response assessment and prognostic evaluation [38–40].

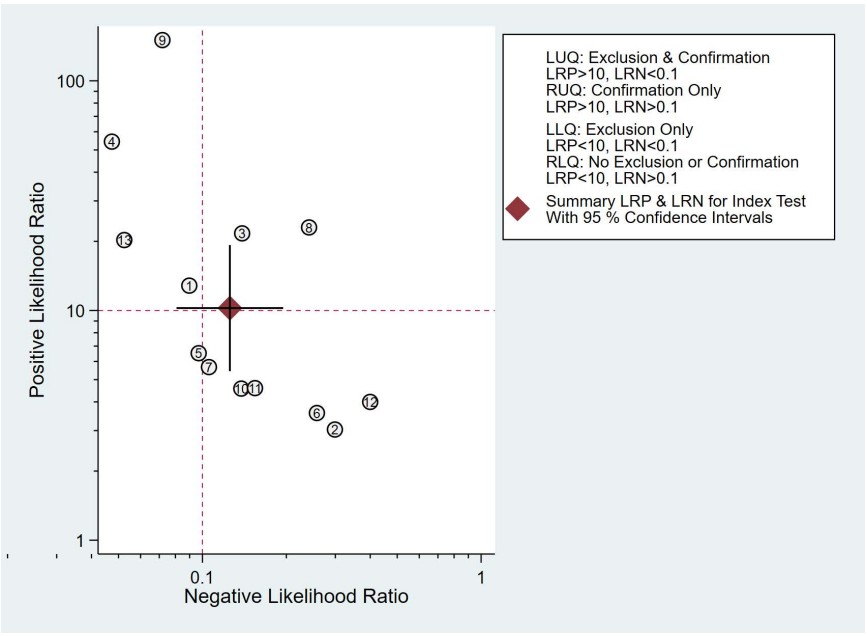

**Fig 5. Likelihood ratio quadrant distribution plot assessing the predictive performance of CA125 for miscarriage risk in patients with threatened abortion.**

Emerging evidence indicates that while CA125 levels remain low during normal pregnancy, they become significantly elevated in certain pathological conditions such as threatened abortion and ectopic pregnancy [23,41–43]. Under physiological conditions, CA125 is abundantly expressed on decidual cell membranes and is primarily secreted into the amniotic cavity, with minimal transfer across the placental barrier resulting in low maternal serum concentrations. However, mechanical disruption of the decidua-placental interface leads to decidual cell damage and subsequent CA125 release into maternal circulation. High levels of CA125 may reflect the inflammatory state of the endometrium, the degree of destruction of the meconium or trophoblast, and the amount of subchorionic hemorrhage, thereby predicting the risk of miscarriage [14,44].

This study has the following limitations. First, the high heterogeneity in specificity (I²=81.82%) suggests that the predictive performance of CA125 for miscarriage risk may vary significantly across studies. Meta-regression analysis indicated that population distribution and sample size were the primary contributors to this heterogeneity. The included studies predominantly involved African populations (61.6%), with European and Asian populations accounting for 26.8% and 11.6%, respectively. The substantial regional disparities imply differences in baseline characteristics, clinical manifestations, management protocols for threatened miscarriage, and detection methods. For example, the gestational age range was broader in African populations (5–19+6 weeks) compared to European and Asian populations (5–12 weeks). Progesterone therapy was partially administered in African populations but not in European or Asian populations. Chemiluminescence assays were predominantly used in African studies, whereas radioimmunoassays were more common in European studies. Those factors may introduce potential selection bias, limiting the generalizability of the findings. Future high-quality, multicenter studies are needed to validate these results. Second, incomplete data reporting in some original studies led to their exclusion, which may introduce bias and affect the accuracy of our conclusions [45,46].

In addition, while our threshold effect analysis suggested pooling was appropriate, the variation in cut-off values across studies should still be considered. European studies reported higher optimal CA125 cut-off ranges (Europe: 43–125 IU/mL,

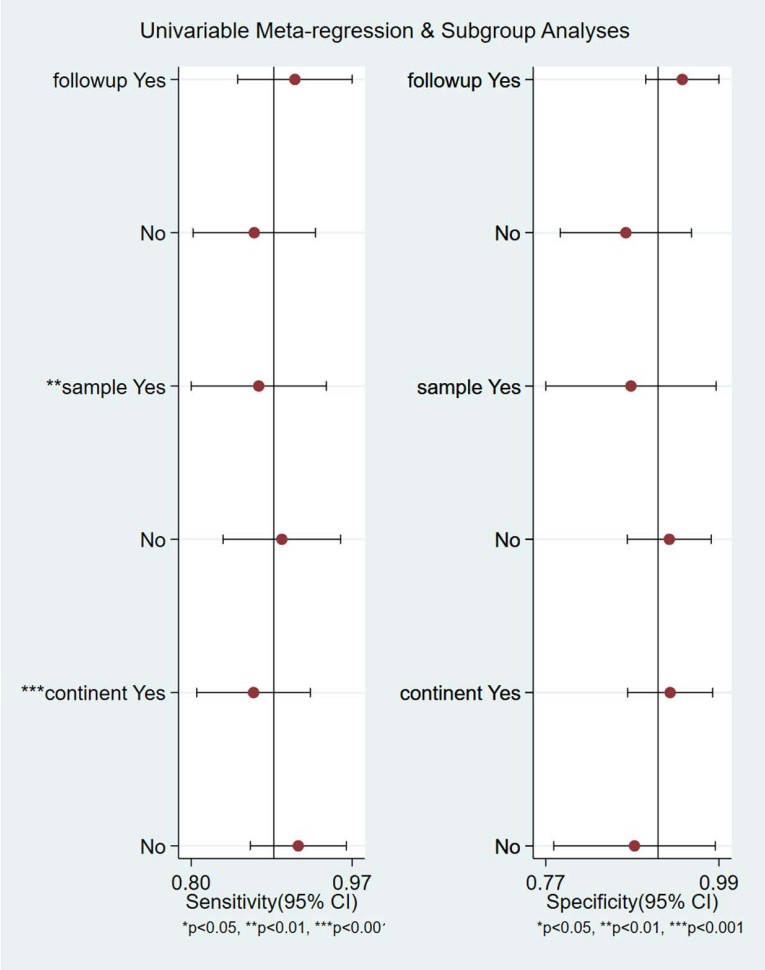

**Fig 6. Meta-regression analysis identifying potential sources of heterogeneity in studies assessing CA125 for miscarriage risk prediction.**

Africa: 21–80 IU/mL, Asia: 55 U/mL), potentially reflecting differences in disease severity, biomarker kinetics, or risk stratification practices. Recent studies have proposed gestational age-specific cut-off values: for patients presenting with abdominal pain alone, 38.25 U/mL at 5 weeks and 53 U/mL at 10 weeks; for those with light vaginal bleeding, 45.6 U/mL at 5 weeks and 68 U/mL at 10 weeks, with values above these thresholds indicating high miscarriage risk [47–49]. Based on current evidence, the substantial variation in cut-off thresholds (21–125 IU/mL) may lead to clinical misclassification in patient stratification, potentially resulting in either delayed intervention or overtreatment. Consequently, a universal threshold cannot be definitively established at this time. Therefore, clinicians should consider local validation of region-specific cut-off values before adopting CA125 as a standalone prognostic tool. In resource-limited settings, CA125 may serve as a simple tool to rule out miscarriage, though ultrasound remains the gold standard. Finally, due to the lack of long-term follow-up data, we were unable to assess the relationship between CA125 levels and long-term outcomes.

In conclusion, as a simple, rapid, and cost-effective biomarker, CA125 demonstrates considerable potential for clinical application in threatened abortion management. Serial CA125 monitoring enables early identification of high-risk pregnancies, facilitating timely interventions such as activity restriction and pharmacotherapy to reduce miscarriage risk. Future studies should employ standardized cut-off values and prospective designs to validate the predictive value of CA125 for

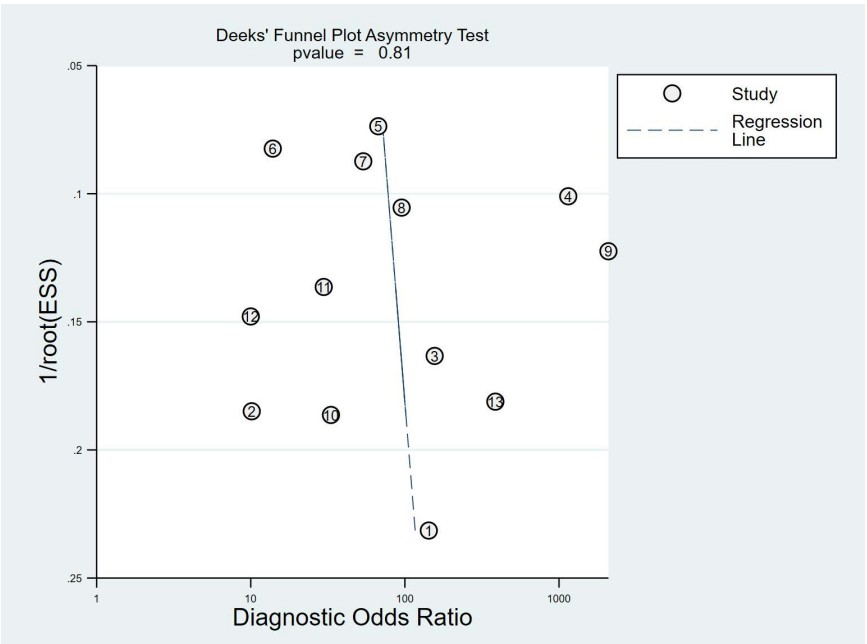

**Fig 7. Deek's funnel plot assessing publication bias in the meta-analysis of CA125 in predicting miscarriage risk in patients with threatened abortion.**

**Table 2. Summary of Findings and GRADE Certainty of evidence assessment: Prognostic accuracy of CA125 for miscarriage risk prediction in threatened abortion.**

| Outcome | Pooled Estimate (95% CI) | GRADE Certainty | Factor of downgrade | Factor of escalation | Clinical Implications |
|---------|--------------------------|-----------------|---------------------|----------------------|-----------------------|
| SEN | 89% (83–93%) | ⊕⊕⊕◯ (Moderate) | Risk of Bias: Participants bias (QUAPAS); Inconsistency: I²=56.43%>50% | Large Effect Size: 89%>80%; Dose-Response: Positive correlation | The false-negative rate is low: approximately 11% of miscarriage cases may be missed by CA125 testing. |
| SPE | 91% (84–95%) | ⊕⊕◯◯ (Low) | Risk of Bias: Participants bias (QUAPAS); Inconsistency: I²=81.82%>50% | Large Effect Size: 91%>80%; | The false-positive rate is low: around 9% of viable pregnancies may be erroneously classified as high-risk. |
| PLR | 10.2 (5.4–19.3) | ⊕⊕◯◯ (Low) | Risk of Bias: Participants bias (QUAPAS); Inconsistency: Wide95% CI (5.4–19.3); Indirectness: Most of these study populations were African, different examination methods and follow-up deadlines | Large Effect Size: PLR>10 | A PLR>10 indicates that a positive CA125 result substantially increases the probability of miscarriage, though the certainty of evidence is low. |
| NLR | 0.13 (0.08–0.19) | ⊕⊕◯◯ (Low) | Risk of Bias: Participants bias (QUAPAS); Imprecision: The lower limit of 95% CI was 0.08 | Large Effect Size: NLR<0.15 | An NLR<0.15 suggests that a negative CA125 result provides moderate utility for ruling out miscarriage, albeit with low-certainty evidence. |
| DOR | 82 (31–212) | ⊕⊕◯◯ (Low) | Risk of Bias: Participants bias (QUAPAS); Inconsistency: Extremely wide 95% CI (31–212); | Large Effect Size: DOR>50 | The high DOR with wide confidence intervals (31–212) implies unstable predictive value across populations. |
| AUC | 0.95 (0.92–0.96) | ⊕⊕◯◯ (Low) | Risk of Bias: Participants bias (QUAPAS); Indirectness: Most of these study populations were African, different examination methods and follow-up deadlines. | Large Effect Size: AUC>0.9 | While the overall discriminative ability is excellent (AUC 0.95), the evidence certainty remains low due to threshold variability. |

Note: GRADE certainty symbols: ⊕⊕⊕⊕=high, ⊕⊕⊕◯=moderate, ⊕⊕◯◯=low, and ⊕◯◯◯=very low certainty of evidence.

miscarriage risk. Furthermore, the diagnostic potential of combining CA125 with other biomarkers warrants further investigation to improve prognostic accuracy and specificity. Additionally, longitudinal follow-up studies are needed to evaluate the association between CA125 levels and long-term pregnancy outcomes, thereby providing stronger evidence for clinical decision-making.

## Supporting information

**S1 Table. Supplementary information of** Table 1**.**
(XLSX)

**S1 File. Raw data and the packages used in STATA program.**
(DOCX)

**S2 File. Excluded articles and reasons.**
(XLSX)

**S3 File. Judgment rationale behind the use of QUAPAS tool.**
(XLSX)

## Acknowledgments

We would like to thank Professor Jun Zhai for his valuable suggestions on this article and Dr. Hao Shi for his support in statistical analysis. We sincerely thank Professor Ahmed Maged for his professional editorial support. His meticulous review, prompt responses, and constructive feedback were invaluable.

## Author contributions

**Conceptualization:** Yurong Cao.

**Data curation:** Yurong Cao, Weiwei Li, Linna Ma.

**Formal analysis:** Yurong Cao.

**Investigation:** Yurong Cao, Weiwei Li, Linna Ma.

**Methodology:** Linna Ma.

**Software:** Yurong Cao.

**Writing – original draft:** Yurong Cao, Weiwei Li.

**Writing – review & editing:** Yurong Cao, Weiwei Li.

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
