## [Decision Letter · Decision Letter 0]

Dear Dr. Yurong,

Thank you for submitting your manuscript to PLOS ONE. After careful consideration, we feel that it has merit but does not fully meet PLOS ONE’s publication criteria as it currently stands. Therefore, we invite you to submit a revised version of the manuscript that addresses the points raised during the review process.

We look forward to receiving your revised manuscript.

Kind regards,

Ahmed Mohamed Maged, MD

Academic Editor

PLOS ONE

Journal Requirements:

2. We note that your Data Availability Statement is currently as follows: All relevant data are within the manuscript and in Supporting Information files.

3. Please include captions for your Supporting Information files at the end of your manuscript, and update any in-text citations to match accordingly. Please see our Supporting Information guidelines for more information: http://journals.plos.org/plosone/s/supporting-information .

4. As required by our policy on Data Availability, please ensure your manuscript or supplementary information includes the following:

Additional Editor Comments:

** Please respond to all reviewers comments**

Reviewers' comments:

Reviewer's Responses to Questions

**Comments to the Author**

1. Is the manuscript technically sound, and do the data support the conclusions?

Reviewer #1: Yes

Reviewer #2: Yes

2. Has the statistical analysis been performed appropriately and rigorously?

Reviewer #1: Yes

Reviewer #2: Yes

3. Have the authors made all data underlying the findings in their manuscript fully available?

Reviewer #1: No

Reviewer #2: No

4. Is the manuscript presented in an intelligible fashion and written in standard English?

Reviewer #1: Yes

Reviewer #2: Yes

Reviewer #1: I thank the respected editor and authors for the chance to review this interesting manuscript. The authors have aimed to investigate the value of CA125 in the prediction of the outcome of threatened abortion and have reported a possible prognostic value. There are a few major and minor points I would like to address:

Major points:

1. Considering that the authors have aimed at reviewing the predictive/prognostic value of CA125 for future events, if authors and editors see fit, the title of the manuscript could be changed to “Assessment of the prognostic value of CA125 for …”. A few sentencing changes should be made to the manuscript text to reflect this change. Keywords could also be adjusted.

2. Accordingly, using the QUAPAS (Quality Assessment of Prognostic Accuracy Studies) risk of bias assessment tool would be better suited in this manuscript, considering that this tool is designed explicitly for prognostic assessment studies while QUADAS tool was designed for diagnostic assessment studies.

QUAPAS Tool: Lee, Jenny, et al. "QUAPAS: an adaptation of the QUADAS-2 tool to assess prognostic accuracy studies." Annals of internal medicine 175.7 (2022): 1010-1018.

In addition, currently, authors have provided no explanations for the reasoning behind giving a judgment of high or unclear risk of bias. I suggest that after utilizing the QUAPAS tool, authors provide the reasonings behind their judgments of high or unclear risk of biases in any of the studies.

3. The authors have mentioned that “Diagnosis was assessed by combining SEN, SPE, PLR, NLR, DOR and AUC.”. Please further explain the analytic methodology, whether true positive, true negative, false positive, false negative values were extracted from articles or meta-analysis was somehow performed with SEN, SPE, PLR, NLR, DOR, and AUC values. Additionally, I suggest authors mention the packages used in STATA program, if any.

4. The utilized thresholds in studies are different. How was this issue addressed in the analysis, or were studies pooled together? If authors have pooled all analysis without considering the utilized cut-off, this could be more clearly mentioned in the methods or limitations section.

5. The authors could also add a summary of findings table and GRADE certainty of evidence assessment (Cochrane Systematic Review Handbook, Chapter 14: Completing ‘Summary of findings’ tables and grading the certainty of the evidence).

Minor comments:

6. In general, the manuscript could benefit from professional English editing, as writing problems and incoherencies, especially regarding verb tense, currently exist.

7. In the methods section, authors have mentioned both Medline and PubMed as databases. Please either remove one of these databases or amend the sentence to something like: “Medline via PubMed”.

8. The authors could cite the two articles that could not be retrieved.

9. Table 1:

The utilized assay in each study for detecting CA125 could be added to Table 1.

Please use one form for synonymous words in Table 1 (venipuncture = venous blood sample).

10. Figure captions could use improvements; in their current form, they aren’t clear as standalone sentences.

e.g: Fig3 could be updated to: “Summary receiver operating curve and empirical bayes estimates for the value of CA-125 in pregnancy outcome prediction in women with threatened abortion”

Please update figure captions to be clear as a standalone sentence.

12. Currently, the raw data necessary for duplicating the analysis are not presented in the manuscript, in contrast to the data availability statement. Please either provide raw data checklist or change the data availability statement.

13. Line 171. A verb seems to be missing after “remaining 22 were.”

Reviewer #2: Dear Authors thank you, please check the attached file

Dear Editor,

Thank you for the opportunity to review the manuscript. This is an updated systematic from “Role of serum biomarkers in the prediction of outcome in women with threatened miscarriage: a systematic review and diagnostic accuracy meta-analysis”. I'm not really sure what the clinical value of this is, some of the articles don't have precise inclusion and exclusion criteria. This factor is influenced by many items, such as VF, which is not mentioned in any of the articles. I have some comments:

Title

• The term “pregnancy outcomes” is broad and could mean different things, such as miscarriage, preterm labor, or live birth rates. A more specific definition of the outcomes being assessed would make the title clearer.

Abstract

• The phrase "clinical value of serum CA125 levels" is not very precise. It would be assisted by specifying whether the focus is on the prediction of miscarriage, pregnancy outcomes in general, or a generalized diagnostic strategy.

• There is no discussion of why CA125 is being evaluated or its significance as a biomarker in threatened abortion. Briefly describing CA125's role in this context would improve accessibility for readers unfamiliar with the topic.

• The conclusion claims “high diagnostic value” for CA125 but does not link this to clinical decision-making or how it could change current practice. Adding a practical implication would strengthen the conclusion.

Introduction

• While CA125 is introduced as a tumor marker, little is said about its physiological role in pregnancy and how it might be associated with risk of miscarriage. The reader is left to wonder why it is being measured.

• Please explain the physiologic and pathologic items that may rise ca125 in pregnancy.

Method

• While the inclusion criteria for threatened abortion are well-described, they could benefit from referencing standardized definitions or guidelines, as variability in diagnostic criteria could introduce inconsistency.

• Whilst heterogeneity due to threshold and non-threshold effects is accounted for, there is no mention of how other sources of heterogeneity, such as study design and population differences, are dealt with.

Result

• he sensitivity and specificity results show significant heterogeneity (I²=56.43% for sensitivity, I²=81.82% for specificity). Although heterogeneity sources are explored via meta-regression, the discussion of how this impacts the reliability of pooled estimates is lacking.

• Although patients are from three continents, the specific proportions from each region are not detailed. Given that the meta-regression shows population differences significantly affect heterogeneity, this could undermine the generalizability of results. 8/13 of the articles are from Egypt. The results may not be very generalizable.

Discussion

• Whereas the discussion indicates that most of the study population was African, it does not provide adequate context for how this unbalance might affect the results' generalizability. An elaboration on demographic differences would be enriching.

• While the manuscript notes variability in CA125 threshold values, it fails to address how this impacts clinical decision-making or propose solutions for standardization.

**Do you want your identity to be public for this peer review?** For information about this choice, including consent withdrawal, please see our Privacy Policy

Reviewer #1: No

Reviewer #2: **Yes: ** Marjan Ghaemi

---

## [Author Response · Author response to Decision Letter 1]

23 May 2025

We sincerely appreciate the reviewers' insightful comments and constructive suggestions, which have significantly improved the quality of our manuscript. We provide a point-by-point response to each comment, detailing the revisions made to the manuscript.

---

## [Editor Report · Decision Letter 1]

Assessment of the prognostic value of CA125 for miscarriage risk in patients with threatened abortion: a systematic review and meta-analysis

PONE-D-24-52697R1

Dear Dr. Yurong,

We’re pleased to inform you that your manuscript has been judged scientifically suitable for publication and will be formally accepted for publication once it meets all outstanding technical requirements.

Kind regards,

Ahmed Mohamed Maged, MD

Academic Editor

PLOS ONE

Additional Editor Comments (optional):

All reviewers comments have been addressed 
---

## [Editor Report · Acceptance letter]

PONE-D-24-52697R1

PLOS ONE

Dear Dr. Yurong,

I'm pleased to inform you that your manuscript has been deemed suitable for publication in PLOS ONE. Congratulations! Your manuscript is now being handed over to our production team.

Kind regards,

on behalf of

Professor Ahmed Mohamed Maged

Academic Editor

PLOS ONE